# Influence of Speech and Cognitive Load on Balance and Timed up and Go

**DOI:** 10.3390/brainsci12081018

**Published:** 2022-07-31

**Authors:** Olivier Van Hove, Romain Pichon, Pauline Pallanca, Ana Maria Cebolla, Sarah Noel, Véronique Feipel, Gaël Deboeck, Bruno Bonnechère

**Affiliations:** 1Department of Pneumology, Erasme Hospital, 1070 Brussels, Belgium; 2Institut de Formation en Pédicurie-Podologie, Ergothérapie Kinésithérapie (IFPEK), 35000 Rennes, France; r.pichon@ifpek.org; 3M2S Laboratory—EA7470, University Rennes 2, 35000 Rennes, France; 4Department of Intensive Care, Erasme Hospital, 1070 Brussels, Belgium; pauline.pallanca@ulb.be; 5Laboratory of Neurophysiology and Movement Biomechanics, Université Libre de Bruxelles, 1070 Brussels, Belgium; ana.maria.cebolla.alvarez@ulb.be (A.M.C.); noux99@hotmail.com (S.N.); 6Laboratory of Functional Anatomy, Université Libre de Bruxelles, 1070 Brussels, Belgium; veronique.feipel@ulb.be; 7Research Unit in Rehabilitation, Université Libre de Bruxelles, 1070 Brussels, Belgium; gael.deboeck@ulb.be; 8REVAL Rehabilitation Research Center, Faculty of Rehabilitation Sciences, University of Hasselt, 3590 Diepenbeek, Belgium; bruno.bonnechere@uhasselt.be; 9Technology-Supported and Data-Driven Rehabilitation, Data Sciences Institute, University of Hasselt, 3590 Diepenbeek, Belgium

**Keywords:** balance, dual task, cognitive loads, Wii balance board, timed up and go

## Abstract

The interaction between oral and/or mental cognitive tasks and postural control and mobility remains unclear. The aim of this study was to analyse the influence of speech production and cognitive load levels on static balance and timed up and go (TUG) during dual-task activities. Thirty healthy young subjects (25 ± 4 years old, 17 women) participated in this study. A control situation and two different cognitive arithmetic tasks were tested: counting backward in increments of 3 and 7 under oral (O) and mental (M) conditions during static balance and the TUG. We evaluated the dual-task cost (DTC) and the effect of speech production (SP) and the level of cognitive load (CL) on these variables. There was a significant increase in the centre of pressure oscillation velocity in static balance when the dual task was performed orally compared to the control situation The DTC was more pronounced for the O than for the M. The SP, but not the CL, had a significant effect on oscillation velocity. There was an increase in TUG associated with the cognitive load, but the mental or oral aspect did not seem to have an influence. Mobility is more affected by SP when the cognitive task is complex. This may be particularly important for the choice of the test and understanding postural control disorders.

## 1. Introduction

The interactions between postural control, motor control and cognitive load have been previously highlighted [1]. However, only a few studies have investigated the impact of the cognitive tasks separately, also referred to as cognitive cost, and the impact of speech production on gait and posture.

Postural control is a complex mechanism resulting from the integration of information from the vestibular, visual and proprioceptive systems [2]. Furthermore, there is a strong link between balance and respiratory functions. Breathing is known to influence balance: different studies have shown that increased tidal volume [3], increased inspiratory load [4] and inspiratory muscle fatigue [5] decrease postural control. On the other hand, cognitive load [6] or the absence of breathing (apnea) seems to improve the balance [7]. This is likely due to a cross effect between cognitive load and breathing, as it has been demonstrated that cognitive load decreases the tidal volume, which can lead to balance improvement [6]. An interesting condition that involves both breathing and cognition is speech. Speaking is characterized by shorter inspiration, longer expiration and an increased tidal volume and respiratory rate [8], leading to deteriorated postural control [9]. Interestingly, postural control and cognitive function are related [10] but the effects of cognition on postural control are still unclear. Indeed, some studies have shown that focusing on postural control could deteriorate balance [11]. Meanwhile, the addition of a cognitive load distraction improves it [12], but the postural sway increases with the difficulty of the task [1]. To summarize, a mild cognitive task improves balance, but postural control is negatively affected if the task is too challenging. The production of speech requires a cortical control of breathing [13]. Therefore, an oral cognitive task may be considered a dual-task activity; this may explain the increased postural sway during vocalization [9].

The motor control required during a mobility task is also influenced by cognitive load. In fact, a motor task may require a high-level process and interfere with cognition [14]. Thus, cognitive performance decreases when walking under a significant cognitive load (i.e., countback in increments of 7) [15]. During TUG, a decrease in mobility and cognitive performance is observed even in young subjects [16]. Controlling breathing for speech could also play a role in increasing cognitive-motor interference.

Speaking while performing a cognitive or motor task can also be considered a double task. Motor aspects of speech are affected by cognitive load. It increases articulatory coordination variability and movement [17]. There is thus a cognitive–motor interaction during speech production. There is also, for example, a complex relationship between hand movements and speech. This relation produces interference, facilitation, or null effects on hand motor tasks depending on conditions and their complexity [18]. For postural control, speaking increases sways [9,19,20] or interferes with the gait in both healthy subjects [21] and patients (e.g., stroke [22]).

Activities of daily living require the ability to perform several tasks simultaneously. It is therefore of the utmost importance to assess the subject’s abilities to perform such complex tasks. In clinics, dual-task tests are the evaluation that mimics the best real-life conditions. These tests allow evaluating the cognitive-motor interference in several circumstances such as walking [23,24] or balance [18,19]. This cognitive-motor interference marks the limit in the ability of humans to manage several tasks simultaneously. This is the dual-tasking paradigm [25]. This suggests that the motor task, which requires attention (more or less important depending on the complexity of the tasks), and the cognitive task share, at least partially, the same brain systems. This implies a decrease in performance in one or both tasks [26]. The decrease in performance observed when adding a task is called the cost of dual tasking [27].

Responding verbally to a cognitive task while performing a task requiring postural control may resemble multitasking. Multiple interferences are possible, including cognitive load on speech, speech on postural control, cognitive load on postural control and postural control on cognitive tasks.

The mental and the oral task will have a different effect on postural control. We hypothesize that the oral task will have a negative effect on postural control, whether dynamic or static. While for the mental task the impact will be less or will improve the postural control. Nevertheless, a mental or oral task may have a different effect on our dynamic static test. This is what we will test in this study and the influence of the cognitive load will also be evaluated. These data could be important when choosing a test to evaluate postural control or establish a dual task training plan. However, also, they can allow a better understanding of balance disorders in the real life. In fact, in real life, we alternate static and dynamic phases as well as oral and mental tasks and light and heavy cognitive tasks. To our best knowledge, the influence of speech on balance and TUG during dual-task activities is still poorly understood. Therefore, the aim of this study was to analyse the influence of speech on static balance and TUG on healthy subjects during different cognitive tasks.

## 2. Materials and Methods

### 2.1. Participants

Thirty healthy participants participated in this study (23.7 ± 2.5 years; 64.9 ± 10.1 kg; 171.6 ± 8.2 cm; 22 ± 2.5 kg/m^2^; 17 women). This study was approved by the Ethical Committee of Erasme Hospital (B4062021000062), and written informed consent was obtained from all subjects prior to their participation. Inclusion criteria were healthy subjects aged 18–40 years Exclusion criteria included neurological conditions, balance deficits or orthopaedic disorders in the last six months.

### 2.2. Protocol

#### 2.2.1. Balance Assessment

Subjects were asked to stand on the middle of the Wii balance board (WBB) (45 × 26.5 cm) as quietly as possible with arms relaxed along the body and fix a target located on a wall two meters away. Participants were asked to not move from the WBB during the protocol to decrease the risk of bias inherent to body position while it has been shown that the position of the foot on the WBB did not influence the results [28]. The WBB is a valid tool to assess balance in different conditions [4,29]. A control situation and two different cognitive arithmetic tasks were tested. Each trial lasted for 60 s, and the order of the five trials was randomly determined. The WBB was connected to a laptop (Intel Core I5, Windows 7, 6 GB RAM) through a Bluetooth connection; data were retrieved using custom-written software based on the WiimoteLib software. The data collection frequency was set at 100 Hz.

#### 2.2.2. Mobility Task

The mobility test consists of a TUG. It is a standardized test commonly used in clinics: the subject is seated on a chair, must stand up, walk 3 m, make a 180-degree turn and sit down again as quickly as possible [30]. The outcome is the time needed to achieve the task. A control situation and two different cognitive situations (see below) were tested.

#### 2.2.3. Cognitive Task

Two different cognitive tasks were evaluated. In the first one, the subjects had to count backwards in increments of 3 (starting from a number randomly selected between 300 and 340, denoted *3*), and in the second, the subjects had to count backwards in increments of 7 (still starting from a randomly selected number between 300 and 340, denoted *7*).

Those two tasks were performed mentally (*M*) and orally (*O*). When performed orally, the ranges (difference between start and end numbers) were computed as well as the number of potential errors. To check that the participants performed the cognitive task under the mental condition, we asked the participants at the end of the mental tasks the final number they reached to assess the range and evaluate if the task was performed error-free.

For reference values, and to be as close as possible to the conditions of balance and mobility, we performed the countback 3 and 7 tests at rest in a seated position: 1 min for the balance test and 15 s for the TUG.

The order of the different conditions (oral or mental) and cognitive load (3 and 7) were randomly defined to avoid fatigue or familiarization. Subjects have 1 min of rest between the different trials and a 10 min wash-out period between the static and the dynamic evaluation. The complete flow of the study is presented in Figure 1.

### 2.3. Data Processing

For the balance assessment, several parameters were computed based on the centre of pressure (COP) displacement using a previously validated method [4]. CP anterior-posterior (CP AP) and mediolateral (CP ML) displacements were obtained from the four strain gauge loads located at the four corners of the WBB using Equations (1) and (2):(1)CPap=(FR+PR)−(FL+PL)
(2)CPml=(FL+FR)−(PL+PR)
where PL, PR, FL and FR are the displacement values from the posterior left, posterior right, front left and front right WBB sensors, respectively. Previous works have shown that the time interval between samples of WBB was inconsistent, therefore, linear interpolation of the raw signals of WBB sensors was applied to obtain a regular sample rate. From those displacements, the nine studied parameters were computed, and descriptions of the computed variables and the equations are presented in Table 1. Data were analysed during the 5th and 55th seconds of each trial, as previous studies have shown that the signal is the most stable during this period.

To evaluate the motor and cognitive interaction, we assessed the dual-task cost (DTC). The different variables and formulas used to assess the influence of the dual task on cognitive and mobility or postural control performance are presented in Table 1. Two cognitive loads (3;7) and two conditions (O; M) were tested. CCR and DTC_mob_ were calculated for 3 and 7 at rest during the quiet standing and TUG. A negative value indicates improvement, while a positive value indicates worse performance.

We then evaluated the effect of speech production (SP) on postural control during quiet standing and TUG. Finally, we evaluated the effect of cognitive load (CL) on postural control during quiet standing and TUG for each condition (oral (CL_O_) or mental (CL_M_)) and for all balance parameters and TUG times.

### 2.4. Statistical Analysis

The normality of each parameter was checked using graphical methods (boxplots, histograms and Q–Q plots) and the homogeneity of variances using the Levene test. As the data were normally distributed, we used the parametric method. Two-way ANOVA was used to compare the effects of the conditions (i.e., control, oral and mental task), the cognitive loads (i.e., 3 and 7) and the interaction between these two factors. Bonferroni’s corrections were adjusted for multiple comparisons in our post hoc analysis. Statistical analyses were performed at an overall significance level of 0.05. Statistics were analysed in RStudio (version 1.2.135) with R version 3.6.1.

To detect a difference of 15% in TMV between the different conditions (for static balance) with 80% power and a two-sided type I error of 5%, we calculated prior to the start of the study the need to include 29 subjects.

## 3. Results

We first discuss the results for the balance, then for the mobility.

First, we compared the results of the cognitive tasks and found no difference between the two conditions (mental or oral) for the simple (3-3 countback) and more complex task (7-7 countback). The results of the balance assessment for the different conditions are presented in Table 2.

We observed a significant effect of the conditions for the ML and AP displacement (RoM) for the speed-related parameters (MLml, MVap, TMV) and for the TUG but no effect of the cognitive loads and no interaction between the conditions and the cognitive loads. Statistically significant mean differences between the conditions with 95% confidence intervals are presented in Figure 2, and complete results and the post hoc analysis results are presented in Appendix A.

Table 3 summarizes the results for dual-task cost mobility and cognitive for the postural control and TUG. We observe a decrease in motor and cognitive performance during the dual task. This decrease in performance seemed to be more marked for the oral task.

The effect of speech production on oscillation velocity and TUG is presented in Table 4. There was a significant effect of speech production on all oscillation speeds. The TUG only increased for countback 7.

There was no significant effect of cognitive load level for the MVml (5 [−18; 29]%, *p* = 0.334), MVap (0 [−25; 26]%, *p* = 0.961) and TMV (1 [−22; 2]%, *p* = 0.842) for the oral condition. We observed the same results on MVml (5 [−11; 21]%, *p* = 0.389), MVap (8 [−13; 29]%, *p* = 0.429) and TMV (6 [−12; 25]%, *p* = 0.411) for the mental condition. The cognitive load level affected the TUG only in the oral condition: CLO (−11 [−26; −13]%, *p* = 0.00025), CLM (−3 [−11; −6]%, *p* = 0.16).

Finally, to compare the static and dynamic (TUG) aspects, we compared the relative changes (in comparison with the control condition) induced by the different cognitive tasks on balance and TUG. We did not find a correlation between these changes for any of the studied conditions (R = 0.20 for Oral 3, 0.36 for Oral 7, 0.08 for Mental 3 and 0.11 for Mental 7, see Figure 3).

## 4. Discussion

The main result of this study is that speech production had a direct influence on gait and posture. First, we evaluated the effect of a cognitive task (countback 3 and 7; oral or mental) on postural control and TUG.

In our research, neither the cognitive task nor its level (cognitive load level) appeared to influence static postural control. Dual tasking is the interference of one task with another when they are performed simultaneously. The effect of a cognitive task depends on its difficulty level. Indeed, a light task decreases postural oscillations [12], whereas a difficult task increases them [1,31]. Our result may indicate that the cognitive task was not sufficiently challenging to produce cognitive-motor interference. The effect of dual tasking on postural velocity illustrates this finding. The impact of dual tasking on sway velocity is comparable across cognitive levels. In spite of this, a recent study demonstrated that, regardless of the level of difficulty, postural oscillation velocities decrease with cognitive load [32]. It should be noted that these studies use different cognitive tasks: arithmetic calculations [32] and executive function on a tablet (i.e., shifting, inhibition, updating) [31]. This disparity in testing can perhaps explain the conflicting results. For dynamic postural control, we used a protocol similar to that of Brustio et al. [16] but we focused on the effect of the dual task on the timed up and go test. The mobility performance (i.e., the dual-task cost) is altered with the cognitive load with a reduction of 20% and 30% for countback 3 and 7, respectively. In contrast to the Brustio study [16], we did not observe any difference in TUG between the levels of cognitive load. However, the dual-task cost on time is 10% higher for countback 7. Other studies have also shown a similar effect of cognitive load level on gait [33]. On the other hand, the cognitive load levels have a negative impact on timed up go in the oral condition. This may be due to a combination of two factors: an increased cognitive load and articulation. It is also known that talking while walking decreases gait speed [16,34]. Subjects seem to prioritise speaking over walking [35]. The difference between countback 3 and 7 may be increased by speaking prioritization.

In this study, we found statistically significant differences between the oral and mental conditions. These results are consistent with those found in the literature on elderly subjects [36].

We found a more significant decrease in postural control with the oral tasks than the mental tasks and control conditions. However, in our study, only the oscillation velocity (i.e., TMV, MVap, MVml) increased, not the COP displacement (DOT, area, AP RoM, ML RoM). The dual task increases the time on TUG; however, there was no difference between the type of tasks (oral vs. mental) and cognitive load (3 vs. 7). Secondly, we analysed the mobility performance (TUG), postural control performance (TMV, MVap, MVml) and cognitive performance during the dual task (dual-task cost). We observed a more significant decrease in postural control performance during the oral task compared to the mental task. An example with the TMV showed a score of 8% for the mental countback 3 and 40% for oral countback 3. As a reminder, the more positive the score, the more the performance is altered. We observed the same thing for cognitive load 7. The TUG evolved similarly but with less marked differences between the oral and mental tasks. The cognitive performance during the dual task in quiet standing was only slightly impaired. This difference was more marked during the TUG. During this dual task, the cognitive performance was more impaired with cognitive load 7 than 3 (67% vs. 4%). Third, to refine our results, we calculated the effect of speech production and cognitive load. Speech production had a significant effect on oscillation velocity. However, this was only observed for countback 7 for the TUG. Nevertheless, these observations further emphasize the importance of the oral task on balance.

The cognitive load level did not affect oscillation velocity but did affect the TUG with countback 7 in the oral condition. These last observations reinforce the effect of speech on postural control.

When comparing our results with the literature, we found that a previous study highlighted that mental tasks could improve postural control [6]; however, when the task is performed orally, there is an increase in postural sway [9,19]. The difficulty of the task also had a negative influence on postural control [1]. In our study, only the oscillation velocity was significantly increased during cognitive tasks in oral conditions. Our results are, therefore, at least partially, in agreement with those of previous studies [9,19,20]. The increase in oscillation velocity can be interpreted as an alteration of balance control [37]. It did not seem to have a difference between the oral tasks (O3 vs. O7) except for mediolateral velocity, nor between the different mental tasks (M3 vs. M7). These observations were corroborated by the effect of speech production. These changes in oscillating velocity during the oral tasks can be attributed to the motor requirements during speech production [19] more than the effect of attention itself. However, other parameters such as the tidal volume changes during speech or mental tasks could also explain this. During a mental task, we found a decrease in tidal volume and a stabilization [6]. Meanwhile, during speech, we found an increase in tidal volume [8], which could induce an increase in oscillations; other parameters such as changes in lung volume could also explain these changes. In this study, we did not assess respiratory parameters and breathing patterns and therefore cannot evaluate these hypotheses. Another hypothesis is that the movements of the jaw will induce modifications in the maintenance of the head. This is important for postural control. Indeed, a stabilization of the cervical spine during a cognitive task (mental) increases postural disturbances [38]. It has been previously demonstrated, for example, that jaw clenching or biting may reduce the postural sway [39]. However, to the authors’ best knowledge, there is currently no study assessing the impact of jaw motion on postural control during speech production. This would be an important factor to study and to determine the implications for the clinics. We did not measure the influence of cognitive load on speaking. However, by modifying the duration of speech-related movements or their variability, it can influence static postural control [17]. This is to be also determined in the future.

In addition, the effect of conducting a dual task on cognitive performance was not modified when the subject was in quiet standing. Cognitive performance was maintained despite the increased cognitive load. It has been previously shown that quiet postural standing [40] and speech production [13] can involve the cortex. Therefore, the task performed in this study could be interpreted as a triple task: cognition, speech production and postural control [21]. However, only postural control was altered. As the cognitive performance was maintained, we hypothesize that, given the possible competitive nature of the cognitive tests, the subjects prioritized the cognitive task over the balance control [41]. We did give any prioritization instructions; however, it could be carried out automatically.

For the TUG, we observed an increase in time associated with the cognitive load, which agrees with a previous study [15]. However, there was no difference in task difficulty or oral or mental type in our study. Nevertheless, the effect of speech production was significant for countback 7, and the effect of the level of the cognitive load was significant during the oral task. This tautologically reinforces our observation of the effect of an oral complex cognitive task on motor performance. This decrease in mobility performance can be explained by the association between the high cognitive task and speech production [21,42]. On the other hand, as previously said, large pre-phonatory breaths involve the premotor cortex [13]. The premotor cortex is also involved in anticipatory postural adjustments during stepping leg selection [43]. A lesion in this area will thus lead to gait dysfunctions. In our study we evaluated healthy subjects, we can therefore hypothesize a competition between speech production and gait control occurring in the premotor cortex.

The multi-tasking effect (speech production, motor task and cognition) was more important for a mobility task than for postural control, which is logical when comparing the tasks’ complexity (balance control vs. TUG). There was a significant impact of mobility on cognition, unlike quiet standing. During the dual task on cognition, the cognitive performance was more impacted for countback 7 than 3 for TUG, which can be considered a marker of a more pronounced interaction between cognition and mobility in the oral condition.

The absence of correlation found between the changes induced by the cognitive loads for the static balance and the TUG (Figure 2) highlights the importance of analysing these two tests individually to have a more precise evaluation of the cognitive-motor interaction.

While these data are important for determining the choice of test type, they are also important for the rehabilitation of these patients. It is important to train the subjects in dual tasks in order to get situations as close as possible to real life. This dual-task training improves walking ability [44] and balance [45] and is more effective than sequential training [46] in elderly subjects with different impairments. However, the exact modalities are yet to be determined [24]. Our study leads to possible research in this area. This study applies to young but the influence of cognitive on postural sways is the same in young and old adults [31]. Multiple tasks, either static or dynamic, are performed in the course of daily life. Or study highlights the possible need to train dynamic and static postural control in a specific manner. In addition, it would be intriguing to observe the results of training with mental and/or oral tasks. In our research, we did not ask the participant to rank the tasks. It may be of interest to investigate the impact of prioritized training (cognitive, motor) as well as the level of difficulty on the subject’s performance improvement [47].

This study has a few limitations, and the results must be interpreted carefully. A first limitation is the absence of tidal volume measurement during the cognitive task. It has been shown that the cognitive task impacts tidal volume and could therefore influence the postural sway [6]. Our observations are, thus, a net result of the oral and cognitive task on postural control without discriminating the effects of the dual task and tidal volume. Another potential limitation is that participants were not given specific instructions on whether they should focus more on the cognitive or motor tasks—this can impact the subjects’ performance, since some may decide to focus more on motor or cognitive strategies [41].

Another limitation of the study is that there is no precise objective assessment of COP during or after the functional task (TUG). The analysis of such variables may bring relevant information into future investigations, particularly in assessing the risk of fall in patients with chronic respiratory diseases [48].

Despite these limitations, we highlighted the complex interaction between speech production, motor aspect and complexity of the cognitive task. Therefore, these factors are important to consider when determining the best tests to assess patients suffering from specific diseases. Further studies need to focus on the impact of various pathologies on these outcomes and relationships. The proposed solution is cost-effective, portable and easy to use and could therefore be easily implemented into daily care.

## 5. Conclusions

In this study, we have shown that the influence of different cognitive tasks on postural control is influenced by the production of speech. During static balance, the oral task seems to significantly alter the balance in healthy subjects more than the similar task performed mentally. However, during a more functional task (TUG), both conditions (oral and mental) had a similar impact on postural control.

This study opens new perspectives for assessing patients with respiratory diseases, cognitive limitations or speaking problems. The results of this study are also of importance for the implementation of specific rehabilitation programs for these patients. Future investigations are needed to confirm our findings and determine the implications of the pathologies in this relationship.

## Figures and Tables

**Figure 1 brainsci-12-01018-f001:**
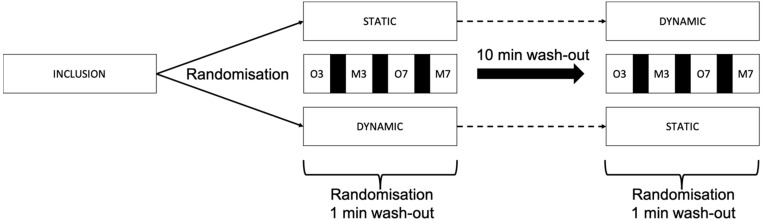
Study flow diagram.

**Figure 2 brainsci-12-01018-f002:**
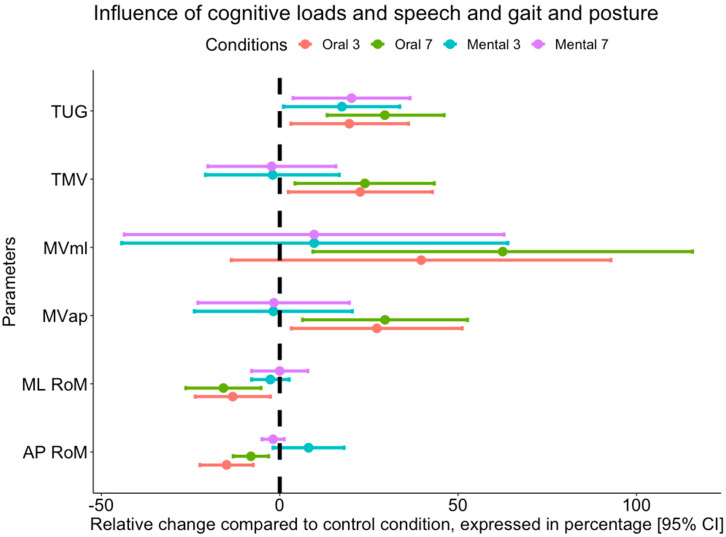
Influence of the different modalities on gait (TUG) and balance-related parameters.

**Figure 3 brainsci-12-01018-f003:**
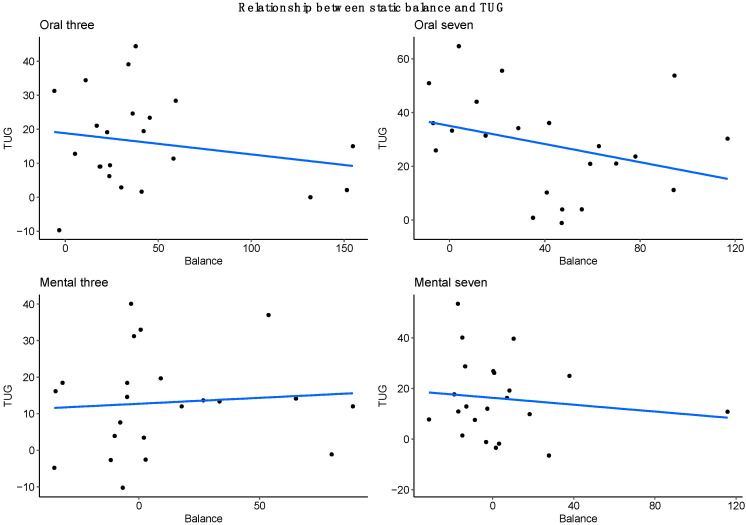
Relative changes (percentage of change relative to the control condition) for the different tasks for TUG and balance.

**Table 1 brainsci-12-01018-t001:** Descriptions of the variables used in this study and equations used to process the data.

Balance		
Name	Description	Equation
DOT	Total displacement of sway	$\sum_{i=1}^{N}\sqrt{CPap{\left(i\right)}^{2}+CPml{\left(i\right)}^{2}}$
Area	The area of the 95% prediction ellipse (often referred to as the 95% confidence ellipse)	$\pi \times prod\left(2.4478\times \sqrt{svd\left(eig\left(cov\left(CPap,CPml\right)\right)\right)}\right)$
AP RoM	The distance between the maximum and minimum COP displacement in the antero-posterior direction	max(CPap)−min(CPap)
ML RoM	The distance between the maximum and minimum COP displacement in the medio lateral direction	max(CPml)−min(CPml)
AP SD	The dispersion of COP displacement from the mean position in the antero-posterior direction	$\frac{1}{N}\sqrt{\sum_{i=1}^{N}CPap{\left(i\right)}^{2}}$
ML SD	The dispersion of COP displacement from the mean position in the medio-lateral direction	$\frac{1}{N}\sqrt{\sum_{i=1}^{N}CPml{\left(i\right)}^{2}}$
AP velocity	The mean AP velocity of COP displacement	$\frac{f}{N}\;\sum_{i=1}^{N-1}\left|CPap\left(i+1\right)-CPap\left(i\right)\right|$
ML velocity	The mean ML velocity of COP displacement	$\frac{f}{N}\;\sum_{i=1}^{N-1}\left|CPap\left(i+1\right)-CPap\left(i\right)\right|$
TMV	The AP and ML displacements of the total COP sway divided by the total duration of the trial	$\frac{f}{N}\;\sum_{i=1}^{N-1}\sqrt{{\left(CPap\left(i+1\right)-CPap\left(i\right)\right)}^{2}+{\left(CPml\left(i+1\right)-CPml\left(i\right)\right)}^{2}}$
**Motor and Cognitive Interaction**		
**Name**	**Description**	**Equation**
CCR	Correct response rate	response rate per second × percent of accuracy
DTC_cogn_	Dual-task cost cognitive expressed in percent. A negative value indicates improvement, while a positive value indicates worse performance.	(single CCR score—dual_task CCR scoresingle CCR score )×100
DTC_mob_	Dual-task cost mobility in percent. A negative value indicates improvement, while a positive value indicates worse performance.	(dual_task mobility score—single_task mobility scoresingle_task mobility score )×100
SP	The effect of speech production on postural control	$DTC_{Mob}\;\left(Mental\right)-DTC_{Mob}\;\left(Oral\right)$
CL_O_	The effect of cognitive load level on postural control during oral tasks	$DTC_{Mob}\;\left(O3\right)-DTC_{Mob}\;\left(O7\right)$
CL_M_	The effect of cognitive load level on postural control during mental tasks	$DTC_{Mob}\;\left(M3\right)-DTC_{Mob}\;\left(M7\right)$

**Table 2 brainsci-12-01018-t002:** Mean (std) results for the studied parameters under the five different conditions. *p*-values are the results of the ANOVA.

Variables	Control	Oral 3	Mental 3	Oral 7	Mental 7	*p*-Values		
						Cond.	Cogn.	Inter.
DOT (mm)	1303 (438)	1109 (324)	1171 (464)	1195 (462)	1199 (521)	0.074	0.92	0.16
Area (mm²)	3488 (2956)	2775 (3120)	2476 (3552)	3472 (4700)	2825 (2963)	0.078	0.72	0.63
ML RoM (mm)	38 (19)	33 (30)	32 (21)	37 (28)	38 (20)	0.021	0.53	0.93
AP RoM (mm)	161 (91)	137 (96)	148 (74)	174 (121)	158 (83)	0.029	0.87	0.32
ML SD (mm)	5.6 (2.7)	5.3 (3.9)	5.4 (4.5)	5.4 (5.0)	6.4 (3.2)	0.056	0.54	0.99
AP SD (mm)	30.4 (17.9)	25.4 (12.8)	27.4 (19.9)	30.8 (21.9)	26.1 (14.7)	0.081	0.99	0.12
MVml (mm/s)	2.8 (0.5)	3.1 (0.6)	2.7 (0.5)	3.0 (0.6)	2.7 (0.4)	<0.001	0.34	0.38
MVap (mm/s)	6.1 (1.3)	7.8 (1.9)	6.0 (1.4)	7.9 (2.5)	6.0 (1.3)	<0.001	0.52	0.21
TMV (mm/s)	7.50 (1.53)	11.2 (3.70)	9.19 (6.35)	12.0 (5.95)	8.00 (2.90)	<0.001	0.82	0.19
TUG, s	4.82 (0.62)	5.77 (1.03)	5.66 (1.21)	6.25 (1.22)	5.80 (1.32)	<0.001	0.43	0.21

Cond. = conditions (control, oral and mental), Cogn. = cognition (3 or 7), Inter. = interaction between conditions and cognitions.

**Table 3 brainsci-12-01018-t003:** Dual-task costs for the different studied parameters. Mean [95% CI].

Balance			
Parameters	Conditions	Oral	Mental
Cognitive	3	4 [−19; 3]%	/
	7	6 [−25; 37]%	/
MVml	3	30 [14; 47]%	6 [−7; 18]%
	7	25 [8; 42]%	0 [−8; 9]%
MVap	3	44 [24; 64]%	9 [−6; 25]%
	7	44 [28; 60]%	1 [−12; 15]%
TMV	3	40 [23; 57]%	8 [−5; 22]%
	7	39 [25; 54]%	2 [−10; 14]%
**TUG**			
Cognitive	3	4 [−12; 19]%	/
	7	67 [59; 75]%	/
Time (mobility)	3	20 [14; 26]%	17 [11; 23]%
	7	30 [23; 36]%	20 [13; 26]%

DTC, dual-task cost; cogn, cognitive; mob, mobility; O, oral; M, mental; 3, countback 3; 7 countback 7, MVml; mean medio-lateral velocity, MVap; mean antero-posterior velocity, TMV; total mean velocity.

**Table 4 brainsci-12-01018-t004:** Mean difference [95% CI] for speech production effect and cognitive load level on oscillation velocity and TUG. *p*-value are the results of the comparison with control situation (paired *t*-test).

Balance					
Effect of Speech Production	Diff.	*p*-Value	Cognitive Load Level	Diff.	*p*-Value
SP3 MVml	−25 [−45; −3]%	0.0001	CLO MVml	5 [−18; 29]%	0.334
SP3 MVap	−35 [−60; −10]%	0.0002	CLO MVap	−0 [−25; 26]%	0.961
SP3 TMV	−32 [−54; −9]%	0.0001	CLO TMV	1 [−22; 2]%	0.842
SP7 MVml	−25 [−44; −17]%	0.003	CLM MVml	5 [−11; 21]%	0.389
SP7 MVap	−43 [−64; −21]%	3.19 × 10^−5^	CLM MVap	8 [−13; 29]%	0.429
SP7 TMV	−37 [−56; −17]%	4.69 × 10^−5^	CLM TMV	6 [−12; 25]%	0.411
**TUG**					
SP3	3 [−6; 11]%	0.343	CLO	−11 [−26; −13]%	0.00025
SP7	10 [1; 20]%	0.0014	CLM	−3 [−11; −6]%	0.16

SP: effect of speech production, 3: countback 3, 7: countback 7, MVml: mean velocity medio-lateral, MVap; mean velocity antero-posterior, TMV: total mean velocity, CLO: cognitive load level oral task, CLM: cognitive load level mental task.

## Data Availability

The data that support the findings of this study are available on request from the corresponding author (BB).

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
