# Peer review of "Influence of Speech and Cognitive Load on Balance and Timed up and Go"

_brainsci, 2022, doi:10.3390/brainsci12081018_

Round 1

Reviewer 1 Report

The study focused its attention on the assessment of the interaction between oral and/or mental cognitive tasks and postural control and mobility. Specifically, the authors analysed the influence of speech production and cognitive load levels on static balance and Time Up and Go during dual-task activities. The study was conducted on a healthy population of young subjects.

There was a significant increase in the centre of pressure oscillation velocity in static balance when the dual task was performed orally compared to the control situation and the mental task. The DTC was more pronounced for the oral with respect to the mental task. The speech production, but not the cognitive load, had a significant effect on oscillation velocity. There was an increase in Time Up and Go associated with the cognitive load, but the mental or oral aspect did not seem to have an influence. Mobility is more affected by speech production when the cognitive task is complex.

The study is very interesting and shed new light on the correlation between balance, mobility, speech and cognitive load paving the way for new considerations also in the context of clinical populations.

The manuscript is well written and, despite the complexity of the statistical part, is easy understandable.

Only two minor consideration: in figure 1 the authors should use all the same labels used in the text.

Reviewer 2 Report

Thank you for inviting me to review this paper. While this topic is of interest, in my opinion, there are major issues mostly in the theoretical background supporting this study. While this paper is on dual-tasking under different motor or cognitive conditions, the authors did not mention any scientific background on mobility/cognitive or balance/cognitive dual tasks. 

The impact of speech while dual-tasking was only considered from a "respiratory" point of view.  However, producing language orally involves many other motor or cognitive aspects not mentionned in the paper. Moreover, there were no respiratory outcomes measured in order to support the main hypothesis (: speech produces more balance perturbation because of its impact on breathing). 

The statistical design should be improved, and a 2 way ANOVA (3 vs 7, and Oral vs. mental) does seem more appropriate to determine the "cognitive load effect" vs the "speech effect". Overall, the methods need clarification, please describe the exact instructions given to participants, how were the different types of cognitive errors taken into account...

Round 2

Reviewer 2 Report

Thank you for allowing me to review this manuscript. Unfortunately, I do not believe that the manuscript was sufficiently improved. The conclusion are not supported by the results (no cognitive load effect was detected but the authors conclude otherwise), the methods section is not described precisely enough (feet position on force platform, exact participants instructions, how errors were calculated, statistics used to produce table 4 p-values...), and the discussion should also mention several classical dual-task hypotheses (eg. posture first for mental-task and not for oral-task?).  

Author Response

Dear reviewer,

We would like to thank you again for your valuable comments and for providing us the opportunity to resubmit a revised version of the manuscript.

We further developed the methodology to describe the instructions given to the subjects and how the errors were estimated as well as the test used to compare the values presented in Table 4.

We added more theoretical background about the dual-task paradigm in the introduction and further elaborate the discussion.

The changes have been highlighted in green (those of the previous revision in yellow). 

Yours sincerely,

Bruno Bonnechère